# The Diverse Reticulate Genetic Set-Up of Endangered *Gladiolus palustris* in Southern Germany Has Consequences for the Development of Conservation Strategies

Marcus A. Koch

Center for Organismal Studies (COS), Department of Biodiversity and Plant Systematics, Heidelberg University, 69120 Heidelberg, Germany; marcus.koch@cos.uni-heidelberg.de

**Abstract:** *Gladiolus palustris* (marsh Gladiolus) of wet grasslands is an extremely rare and highly endangered species in Central Europe. Ongoing loss of habitat, population fragmentation, drought, and higher mean annual temperatures caused by global warming have all contributed to a severe decline in its population over the past few decades. Additionally, hybridization with other species, such as *G. imbricatus* and *G. illyricus*, and genetic depletion may pose a significant threat to the species' survival. The focus of this study is to characterize major gene pools of the species in southern and southwestern Germany. Using molecular AFLP markers and ITS DNA sequencing, this study shows that past hybridization and introgression in Central Europe are more extensive than previously thought, posing a challenge to conservation strategies targeting taxonomically defined species. The region of the Rhine River in southwestern Germany (Upper Rhine Valley) has seen the emergence of various scattered populations of *G. palustris* over the past three decades, which are believed to have been introduced by humans. Introduced populations in this area (comprising the German Federal States of Baden–Württemberg and Rhineland–Palatinate) likely descend from a large source population near Lake Constance. Therefore, the study suggests promoting and protecting these new populations, given their long-standing presence in the region. Furthermore, the research proposes that naturally occurring hybrids and introgressed populations should also be the primary target of conservation efforts.

**Keywords:** Central Europe; conservation biology; genetic diversity; gene pools; *Gladiolus palustris*; hybridization; human-mediated introduction



## 1. Introduction

Plant species in Central Europe are more and more endangered, irrespective of whether they are restricted to natural or semi-natural landscapes or if they are closely associated with the cultural landscape that has been shaped by human activities, such as agriculture, forestry, and settlements over the past centuries [1]. The reasons for population decline include land-use intensification, increasing levels of phosphate and nitrogen, changes to the hydrological system, habitat destruction and population isolation, and, during the past decades, climate change, which has had a severe impact on changing temperatures and precipitation regimes [2,3]. Population decline, isolation, and dispersal limitations are severely reducing genetic diversity [4], and as a consequence, genetic drift, inbreeding depression, and loss of adaptive capacity are further contributing to decreases in population fitness and foster population decline [5].

In many cases, environmental change and habitat disturbances also shift the distribution ranges of closely related species and force them into genetic contact [6,7]. In the evolutionary past, this resulted in the formation of hybrid zones and eventually even into new species in Central Europe, e.g., during Pleistocene glaciation and deglaciation cycles [8–10]. However, present-day environmental changes are also severe, and the chance of secondary genetic contact of formerly separated gene pools of related species is therefore

enhanced [11]. As a consequence, hybridization and introgression may tear down species boundaries. Intermediate and competing hybrids may cause a severe threat, which is typically more pronounced in specialized, sensitive, and often rare and endangered species. In Central European flora and in particular in southwestern Germany—the focal area of this study—such examples from vascular plants are the very rare *Anthriscus stenophylla* (genetically eroded by *A. sylvestris*), *Aquilegia vulgaris* (eroded by *Aquilegia* cultivars with North American genetic background), *Nymphaea candida* and *Nymphaea alba* (eroded by *N. alba* and North American Nymphaea cultivars, respectively), or *Nuphar pumila* (eroded by *N. lutea*) [12–14].

Consequently, there is an increasingly important discussion about the outcome of hybridization and introgression among closely related taxa or the merging of different gene pools of the same taxon due to ongoing environmental change and human impact and how to deal with them in legal and conservation frameworks [15,16].

The marsh gladiolus (*Gladiolus palustris*) is an example for such a species, which is potentially threatened by hybridization and introgression. The species is distributed throughout Western and Central Europe in calcareous fens, wet straw meadows, and semi-dry calcareous grasslands but also semi-dry forests on calcareous soils [17,18]. Plants are well-adapted to these environments and are often found in areas adjacent to calcareous fens with spring-fed locations that can be periodically flooded. In the hypothetical natural landscape without anthropogenic influence, moist scree slopes and open forests may have served as original habitat type [17]. The species is highly threatened in many countries and protected according to the Fauna–Flora–Habitat Directive of the European Union (Annex II Habitats Directive 92/43/EEC). Closely related European species with putative or even proven distribution range overlap and genetic contact with *G. palustris* are *G. imbricatus* L. and *G. communis* L. [18,19]. In the case of *G. imbricatus*, a naturally occurring hybrid with *G. palustris* has been recognized as the nothospecies *G.* x *sulistrovicus* [20] in Poland. It has been further substantiated that ancient hybridization processes also gave rise to a few western European populations in France [19].

In Germany, *G. palustris* is restricted to the southern Federal States Rhineland–Palatinate, Baden–Württemberg, and Bavaria. In the first two states the species is largely associated with the Upper Rhine Valley system. In the latter case of Bavaria, present-day distribution is particularly associated with the Bavarian Alpine foothill region. However, only a few large populations have persisted, such as in the Mutterstadt area (Rhineland–Palatinate), in the Wollmatinger Ried nature reserve at Lake Constance (Baden–Württemberg) and the largest European population near Augsburg (Königsbrunner Heide, Bavaria). Over the past decades, a few scattered and very small populations have been newly documented along the Upper Rhine Valley system, and it has remained unclear if these populations (i) were (illegally) introduced by humans and (ii) if they belong either to *G. palustris* regional gene pools, naturally occurring hybrids or even cultivated garden offsprings. Thus, a representative set of populations from Western Europe has been studied using AFLP markers and sequences of the ITS region of the nuclear ribosomal DNA.

In particular, it is asked herein if *G. palustris* gene pools in Germany show severe footprints of past and present introgression and hybridization, and if also newly found populations in Baden–Württemberg serve a status as worthy of protection. The results should also contribute to developing settings of conservation guidelines in respect to species concepts and species boundaries.

## 2. Materials and Methods

### 2.1. Focal Species, Study Area, Sampling, and Experimental Design

*Gladiolus palustris* (marsh gladiolus) is threatened throughout its Central European distribution range, suffering from habitat degradation and loss and increasing fragmentation and isolation of populations [19]. Furthermore, *G. palustris* has been proven to establish naturally occurring hybrids and hybrid taxa. This applies not only for the eastern part of the distribution area of *G. palustris* in Poland and the Czech Republic [18,20] (involving

*G. imbricatus*), but hybrids have also been identified (involving *G. imbricatus* and perhaps *G. communis*) in the western part of its distribution area in France and Switzerland [19]. This study focuses on the genetic status of southern German *Gladiolus* populations, bridging results from earlier analysed geographic regions [19,21]. The study region comprises the German Federal States Baden–Württemberg and Bavaria and the northern part of Austria (Figure 1) and has been originally initiated in collaboration with LUBW (Baden–Württemberg State Institute for the Environment, Measurements, and Nature Conservation) seeking support to evaluate the genetic and conservation status of eventually and illegally human-introduced individuals at various places throughout Baden–Württemberg. For this purpose, leaf material from all relevant populations in Baden–Württemberg had been collected (Table S1). In addition, we are grateful to B. Leitner and A. Tribsch (Salzburg University) for sharing numerous DNA samples from Austria [21]. The focus of this study is to define regional gene pools. It is also investigated if populations introduced by humans show genetic signatures indicating hybridization and introgression. Therefore, single or few individuals per population have been sampled at habitat sites in question. Because of earlier reports of naturally occurring hybrids, we also explored species identity using DNA sequence information from the nuclear encoded internal transcribed spacer regions ITS1 and ITS2 [18]. Four populations investigated herein have been considered in Baden–Württemberg to be illegally introduced by humans (G10, G12, G15, G16; Table S1).

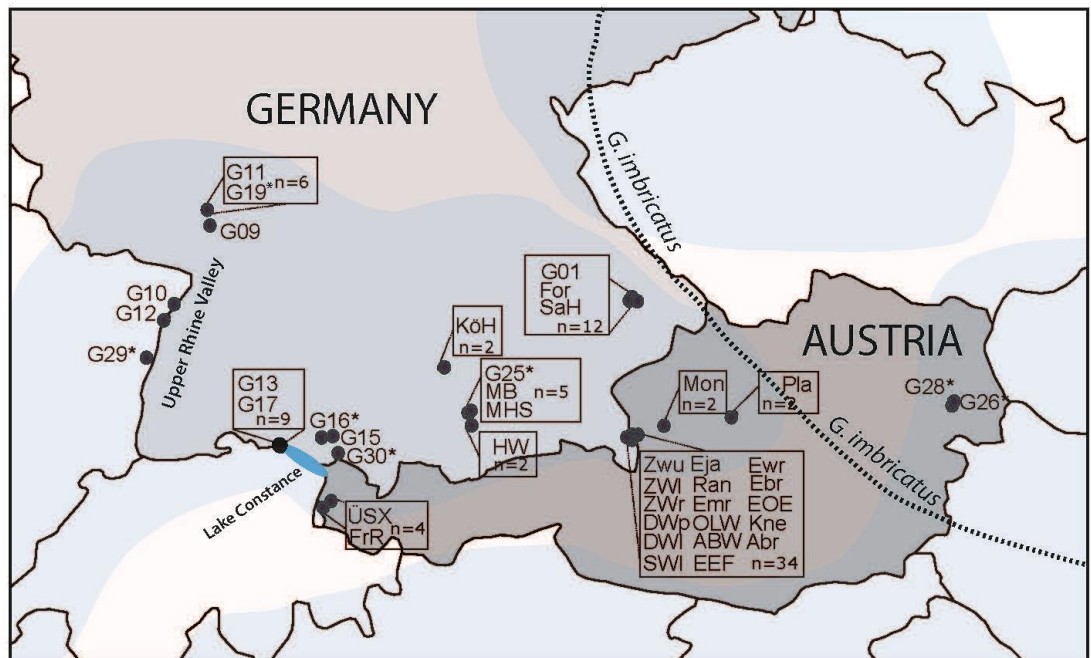

**Figure 1.** Populations of *G. palustris* analysed for genetic assignment using AFLPs. Population code is indicated and more details are found with Table S1. The bluish underlaid area denotes the distribution range of *G. palustris*. The dotted line indicates the western margin of occurrence of *G. imbricatus* (redrawn from [19]). Asterisks mark those populations with individuals showing an unusual placement in SplitsTree analysis, and these individuals have been also studied for ITS sequence variation to analyse species identity. "n" indicates the number of individuals analysed. If "n" is not indicated, a population is represented by a single individual.

The combination of AFLPs and ITS was selected to reliably detect gene pools to compare across earlier studies and also integrate sequence data from GenBank originating from populations across Europe. For AFLP analysis, 42 locations were sampled to represent *G. palustris* (native or introduced populations), among them 3 populations obtained from in situ conservation collections (G01: Deggendorf; G17a and G17b: Wollmatinger Ried). As outgroup accessions, 3 *G. illyricus* samples were selected from Croatia and Portugal.

For additional DNA sequence-based species delimitation, one accession of *G. italicus* from Cyprus is included (Table S1).

### 2.2. AFLP-Based Genotyping and Genetic Data Analyses

A total of 90 individuals of *G. palustris* (Figure 1) representing local populations and 3 individuals of *G. illyricus* (outgroup) were genotyped. Genomic DNA was extracted from 50 mg leaf tissue, which was dried in silica gel during sampling in the field, following a CTAB protocol [22] with minor modifications [23]. The leaf tissue was ground into a powder in a Precellys 24 homogenizer (Bertin Technologie, Montigny-le-Bretonneux, Frances). The extracted DNA was washed twice (70% ethanol), dissolved in 50 µL TE buffer, and supplemented with 2 U of RNAse A. RNAase treatment was performed at 37 °C for 1 h. All samples were adjusted to the same concentration of 100 ng/µL by adding ddH$_2$O after the DNA concentrations were measured (Quibit fluorometer, Thermo Fisher Scientific, Waltham, MA, USA). This DNA was used to perform AFLP analysis [24] using an established protocol [25]. The initial step (restriction ligation reaction) was performed in a total volume of 12 µL, which contained 45 ng genomic DNA, 2 U T4 DNA ligase (NEB, Frankfurt, Germany) in 1 × T4 ligase buffer, 2 U *Mse*I, 2 U *Eco*RI, 75 pmol *Mse*I-adapter pair (5′-GACGATGAGTCCTGAG-3′ and 5′-TACTCAGGACTCAT-3′), and 75 pmol *Eco*RI-adapter pair (5′-CTCGTAGACTGCGTACC-3′ and 5′-AATTGGTACGCAGTC-3′). The restriction ligation (RL) step was followed by digestion of the DNA at 37 °C for 3 h, followed by 10 min. at 65 °C in a thermal cycler. The digestion product was then incubated overnight at 16 °C for adapter ligation. For subsequent pre-selective amplification, RL products were diluted 10-fold with sterile ddH$_2$O and using *Mse*I + C (5′-GACGATGAGTCCTGAGTAAC-3′) and *Eco*RI + A (5′-GACTGCGTACCAATTCA-3′) as the pre-selective primers. The reaction volume was adjusted to 12.5 µL (2.5 µL diluted RL mixture, 1 × PCR buffer, 2.0 mM MgCl$_2$, 200 uM dNTP, 40 pmol of each primer, and 2.5 U Taq DNA polymerase). PCR amplification followed the following scheme: hold at 72 °C for 2 min, followed by 20 cycles of 94 °C for 20 s, 56 °C for 30 s, and 72 °C for 2 min, and a final hold at 72 °C for 60 min. For final selective PCR, the pre-selective products were diluted 10-fold with 1 × sterile ddH$_2$O, and selective PCR amplification was performed using three 5′ fluorescence-labeled primer pairs used in an earlier study [21] to allow for further comparisons among studies: *Eco*RI-ACT (5′-GACTGCGTACCAATTCACT-3′) (FAM)/*Mse*I-CA (5′-GATGAGTCCTGAGTAACT-3′); *Eco*RI-AGG (5′-GACTGCGTACCAATTCAGG-3′) (HEX)/*Mse*I-CT (5′-GATGAGTCCTGAGTAACT-3′), and *Eco*RI-AAC (5′-GACTGCGTACC AATTCAAC-3′) (TAMRA)/*Mse*I-CT (5′-GATGAGTCCTGAGTAACT-3′). A reaction volume of 12.5 µL was prepared for each primer pair (2.5 µL diluted pre-selective products, 1 × PCR buffer, 1.5 mM MgCl$_2$, 300 µM dNTP, 4 pmol of each *Eco*RI fluorescence-labeled primer, 25 pmol *Mse*I primer, and 1 U Taq DNA polymerase). The final amplification used the following protocol: hold at 94 °C for 5 min, 12 cycles of 94 °C for 30 s, 65 °C for 30 s (minus 0.7 °C per cycle), 72 °C for 2 min, followed by 20 cycles of 94 °C for 30 s, 56 °C for 30 s, 72 °C for 2 min, and a final hold at 72 °C for 30 min. The three-selective PCR products were pooled for each sample and genotyped at Eurofins Genomics (Eurofins Genomic GmbH, Ebersberg, Germany). Four samples of *G. palustris* were replicated to further calculate the experimental error rate [26]. Raw data provided by Eurofins were scored using GeneMarker v1.95 (SoftGenetics LLC, State College, PA, USA). The data were then exported as a presence/absence matrix. The scored fragment size was restricted to a range from 60 to 410 nucleotides to further improve the quality of the data set. The AFLP presence/absence matrix was further inspected manually. Genetic distance analysis used the Jaccard distances among samples, which were calculated from the AFLP data using R function [distance(x), method = "jaccard"] [27]. The respective genetic networks were calculated using the neighbour-net algorithm [28] as implemented in SplitsTree4 version 4.15 [29,30].

Genetic assignment of individuals to genetic clusters at the landscape level was conducted using STRUCTURE version 2.3.4 [31]. STRUCUTRE is a model-based Bayesian

clustering method and uses multilocus data to infer population structure and assign individuals to putative (sub)populations and allows for the analyses of dominant data such as AFLP [32]. In this study, there were a limited number of individuals sampled per population analysed, and the species studied has a high vegetative propagation potential. Therefore, both the admixture and no-admixture options were used in different runs without prior information about the regional origin of the populations and assumed that allele frequencies are correlated. In order to study the number of optimal genetic clusters ten runs with 100,000 iterations were carried out after a burn-in of 10,000 iterations to quantify the amount of variation in the posterior probabilities of the data for each cluster number (*K*). A range for *K* between 1 and 10 has been chosen. The mean estimates of the posterior probabilities of the data for a given cluster number L(*K*) and the statistic Δ*K* were used [33] to estimate the number of significant clusters using STRUCTURE Harvester [34]. The various runs of STRUCTURE simulations for the most appropriate *K* were aligned using the Greedy option in CLUMPP v1.1.2 [35].

SplitsTree and STRUCTURE analyses are complemented by a principal component analysis (PCA) to compare among the various results. PCA was performed using ClustVis [36]. Unit variance scaling is applied to rows of data, and singular value decomposition (SVD) with imputation is used to calculate principal components. Predictions ellipses are calculated such that with a probability 0.95, a new observation from the same group will fall inside the ellipse [36].

Non-hierarchical AMOVA with the total variation partitioned among and within the main geographic regions studied (Baden–Württemberg/Rhineland–Palatinate in Germany: largely characterized by the Rhine valley; Bavaria: Bavarian lowlands; and Austria: alpine foothill regions) and gene diversity analysis were performed using ARLEQUIN v. 3.5.2.2 [37]. In addition, genetic diversity parameters (number of rare alleles with a frequency < 0.05, Jaccard distance matrix) were studied for the same groups comparing Baden–Württemberg/Rhineland–Palatinate (group 3), Bavaria (group 2) and Austria (group 1).

*2.3. Species Assignment Using DNA Sequence Information from Internal Transcribed Spacer 1 and 2 (ITS) and Phylogenetic Analyses*

Based on results from STRUCTURE and SplitsTree *G. palustris* individuals from populations G16, G19, G25, G26, G28, G29, and G30 showed a genetic assignment signature indicating genetic contribution from other species than *G. palustris*. All these individuals plus outgroups *G. illyricus* (G22, G23, G24) and *G. italicus* (814618) were analysed for ITS DNA sequence variation. In addition, one Austrian population (ZWU) studied earlier [21] and one individual from population G17 were included as representatives of genetically closely assigned *G. palustris* individuals. The original PCR products and three cloned PCR fragments were sequenced for any of those samples. ITS was chosen because of the availability of numerous sequences from two case studies exploring *G. palustris*, *G. imbricatus* and putative hybrids between both species (KP027306-KP027328 [18], MK005888-MK005919 [19], while the later study did not incorporate previous sequence data. Additional sequences were available from a barcoding study (MF543717-MF543720) [38].

PCR amplification of the ITS region was carried out using the primer combination ITS-M13 (forward) [TGTAAAACGACGGCCAGT-GGAAGGAGAAGTCGTAACAAGG] and ITS-M13 (reverse) [CAGGAAACAGCTATGACC-GGGTAATCCCGCCTGACCTGG]. PCR products spanned the entire ITS1, 5.8 S rDNA, and ITS2 region. The PCR scheme was performed using approximately 50 ng of genomic DNA, 0.5 µL of deoxyribonucleotide triphosphates/dNTPs) mix (10 mM), 1 × MangoTaq buffer, 1 µL MgCl2 (50 mM), 1 U MangoTaq DNA polymerase (Thermo Fischer, Dreieich, Germany) and 0.5 µL of each primer (10 µM), in a total volume of 25 µL. Amplification was performed using the following protocol: hold at 95 °C for 3 min, 29 cycles of 95 °C for 30 s, 44 °C for 30 s, 72 °C for 1 min, followed 72 °C for 5 min, and a final hold at 4 °C. PCR products were purified using the QIAquick PCR Purification Kit Protocol (Qiagen, Hilden, Germany) and cloned into

the CloneJET PCR vector system (Thermo Fischer, Dreieich, Germany). Custom Sanger sequencing was performed at EUROFINS (Ebersberg, Germany) with original PCR products and 3 cloned fragments from any DNA sample selected using the M13 forward and reverse primers.

Raw sequencing ITS data were imported into Geneious 11.1.5 [39] and processed further. The final alignment of the sequences plus sequences retrieved from Genbank was performed with MUSCLE [40] and adjusted manually. The GTR+Gamma model of nucleotide substitution was selected according to ModelFinder [41], and maximum likelihood (ML) analysis was performed using IQ-Tree [42] on the IQ-webserver [43]. The number of bootstrap replicates was set to 1000. Bayesian inference was performed using MrBayes version 3.2.6 [44]. Four simultaneous runs with four chains each were run for 50 million generations, sampling every 1000 trees. The first 25% of these trees were discarded as burn-in when computing the consensus tree (50% majority rule). For efficient swapping of the chains, the temperature of the heated chain was set to 0.005. Sufficient mixing of the chains was considered to be reached when the average standard deviation of split frequencies was below 0.01.

## 3. Results

### 3.1. AFLP Analysis of Gladiolus Palustris Accessions

Using three primer combinations, 313 AFLP marker bands were identified and scored as 0/1 matrix (Table S2). Among these bands, 41 occurred only in one single individual. *Gladiolus illyricus* accessions did not exhibit synapomorphic marker bands. All 90 individuals had a unique multilocus genotype. For AFLP analysis, error rates below 5% are considered adequate [25]. The calculated error rate was 2.25%.

SplitsTree analysis based on Jaccard distances provided good evidence for the genetic separation of *G. illyricus* (Figure 2). However, a number of individuals from populations G25, G26, G28; G16, G19 (individual no. 82), G29, and G30 were set apart from the other *G. palustris* individuals. These individuals originated from populations in France, Austria, and the German Federal States Bavaria, Baden–Württemberg and Rhineland–Palatinate without displaying any obvious geographical pattern and were considered as potential hybrids—either naturally occurring or introduced by humans. STRUCTURE analyses of the same *G. palustris* dataset revealed a similar signal, which is fully consistent with SplitsTree results under the no-admixture option (Figure S1; Table S3a,b). For both analyses (admixture and no-admixture using the total dataset), optimal *K* was 2, which is considered to be caused by the drastic effect of the alien genotypes (Figure S1). In search of less prominent genetic STRUCTURE signal among *G. palustris* populations, the above-listed individuals were excluded, which results in a most optimal *K* = 5 under the no-admixture option (Figure S1, Table S3c,d). Accordingly, regionally structured genetic variation of *G. palustris* aligned using CLUMPP is exemplified in Figure 3 and displays the result using the no-admixture option.

### 3.2. Genetic Variation within and among Regions of Gladiolus palustris Accessions

Because the sample size of the populations is too low to calculate population-based genetic parameters, individual information is combined on a regional level with and without including outlier individuals (Table 1). Inclusion of those outlier individuals increased regional genetic diversity consistently, indicating that those individuals contribute to additional allelic variation, and the estimated amount of regional expected heterozygosity decreased (Table 1). Consistently expected heterozygosity and number of rare alleles are highest in the group 3 populations (Baden–Württemberg and Rhineland–Palatinate). The number of rare alleles differs significantly from values calculated for group 1 (Austria) and group 2 (Bavaria) ($p < 0.01$, for any group comparison, Mann–Whitney U-test). The largest effect of introducing rare alleles into the dataset is seen in population G30 from Eastern Austria, with 21 rare alleles found in the respective individuals, which is fully consistent with SplitsTree analysis (Figure 2) indicating genetic contribution from an unknown genepool.

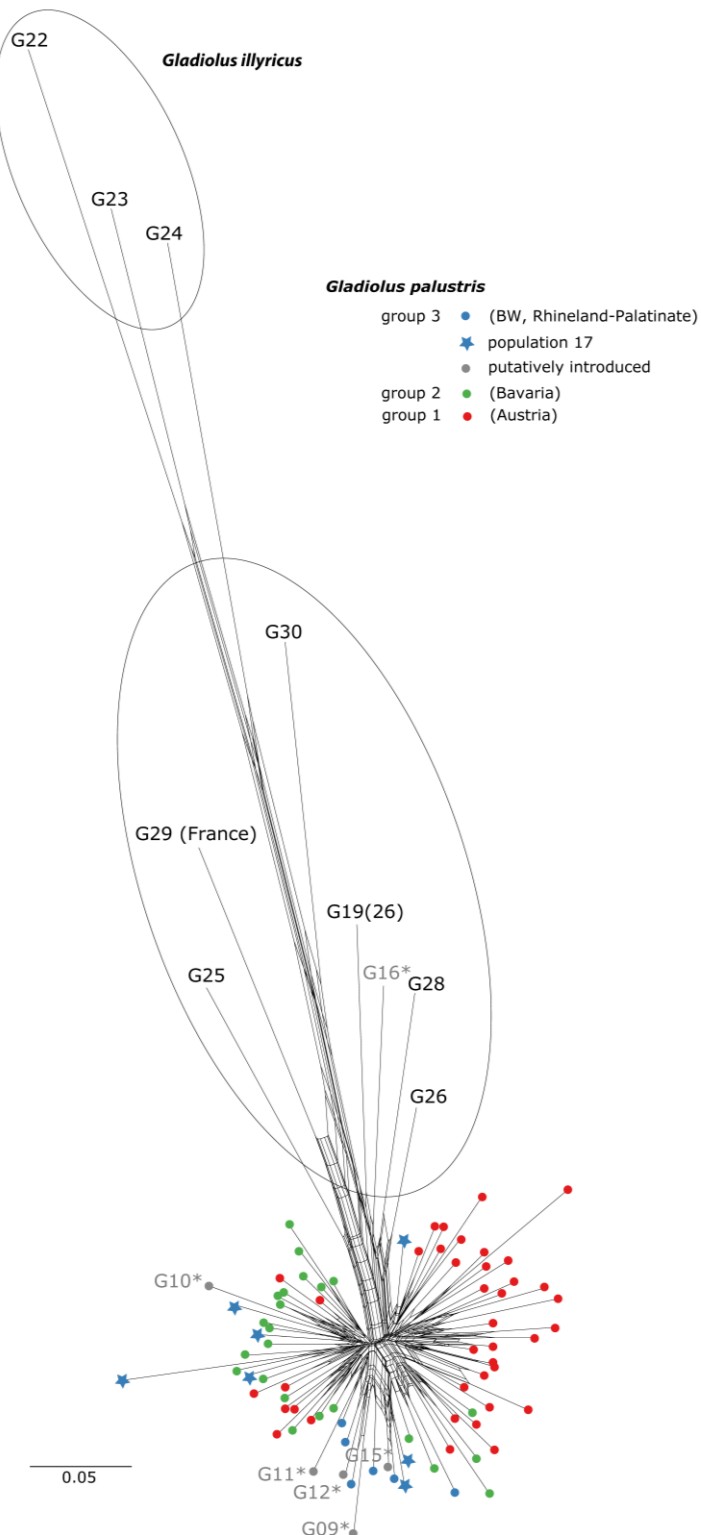

**Figure 2.** SplitsTree graph of AFLP data from *G. palustris* and *G. illyricus* based on pairwise Jaccard distances (scale indicated in Figure). *Gladiolus illyricus* accessions (22, 23, 24) served as outgroup. In total, 87 individuals of *G. palustris* populations (Figure 1) have been included. Putatively introduced populations in Baden–Württemberg (group 3) are indicated in grey (population code and dots) and are marked with an asterisk. Assignment of populations to regionally defined groups (1, 2, and 3) corresponds to Figure 3 (STRUCTURE analysis) and Figure 4 (PCA analysis). The oval in the middle indicates potential hybrid individuals.

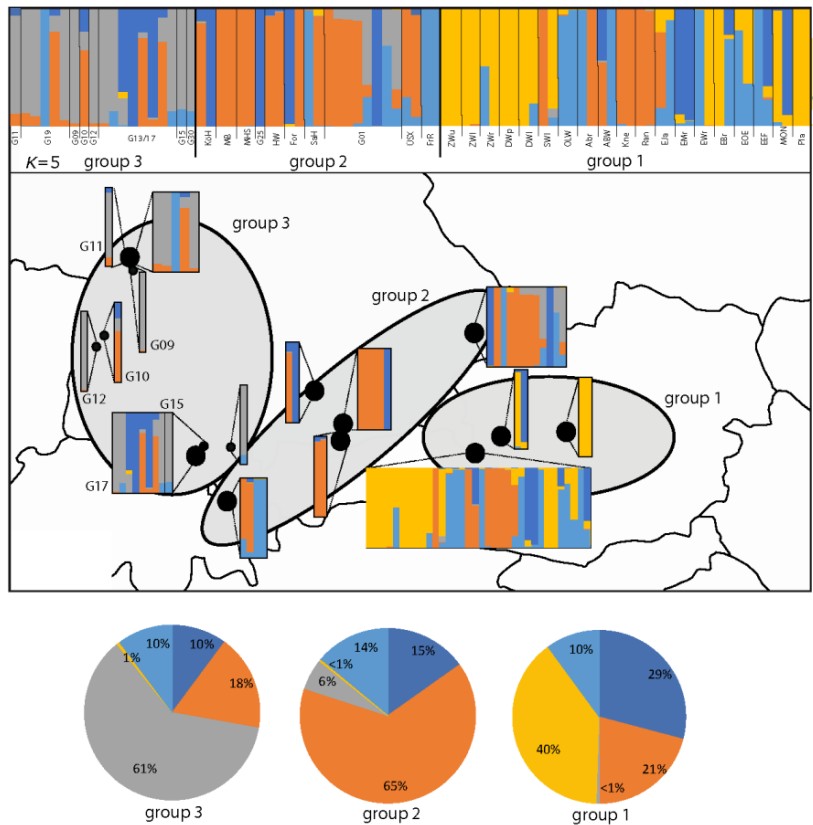

**Figure 3.** Estimated population structure of *G. palustris* as revealed via STRUCTURE analysis. Each individual is represented by a vertical line and is portioned according to optimal *K* = 5. CLUMPP was used to align the runs of ten simulations. The top lines are sorted to separate German and Austrian populations according to groups 1, 2, and 3. The population code is provided accordingly. The map depicts in more detail individuals structured geographically according to three main distribution regions: group 3 (Baden–Württemberg and Rhineland–Palatinate); group 2 (Bavaria); group 1 (Austrian populations). See Figure 1 for further details.

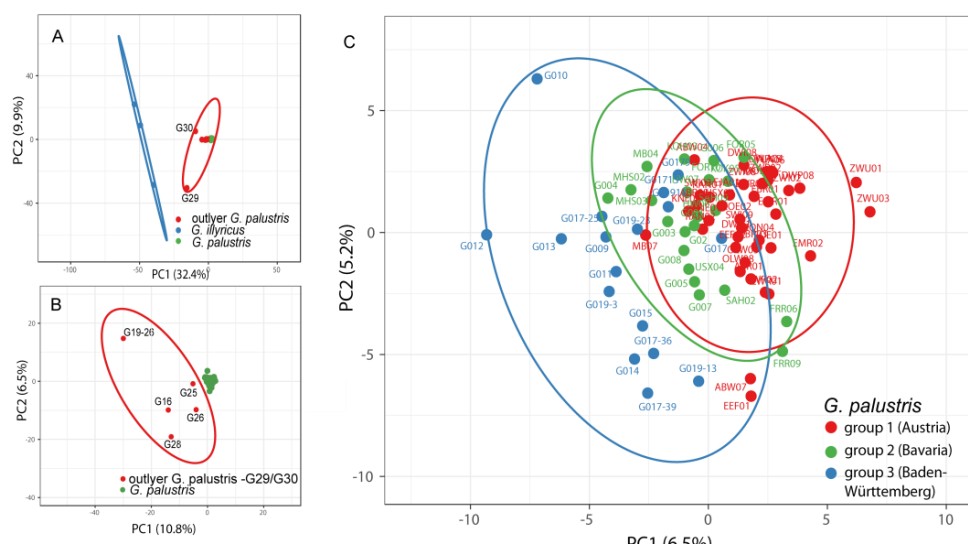

**Figure 4.** Principal component analysis (PCA) of AFLP data from *G. illyricus* and *G. palustris*. *x* and *y* axis show principal component (PC) 1 and 2. (**A**) Total dataset comprising all samples as used in SplitsTree analysis (Figure 2). (**B**) Dataset including all *G. palustris* accessions except outlier populations

G29 and G30. (**C**) Dataset comprising all populations of *G. palustris* excluding the outgroup (*G. illyricus*, G22, G23, G24) and all outlier populations (G16, G19, G25, G26, G28, G29, G30). Colour code in Figure 4C is congruent to colour code used in SplitsTree analysis (Figure 2) to define groups 1, 2, and 3.

**Table 1.** Genetic diversity parameters of the three groups of *G. palustris* population studied (group 1, 2, 3; N = populations; n = individuals) as revealed with Arlequin analysis. Standard deviation is provided in brackets. All analyses were performed with the reduced dataset denoted with * (excluding G16, G25, G26, G29, G30, G19#82) and the total *G. palustris* dataset (non-reduced). Group 1 (Austria), group 2 (Bavaria) and group 3 (BW and Rhineland–Palatinate).

| | N/n | Expected Heterozygosity | Gene Diversity | No. Polymorphic Loci | No. Rare Allels (<5%) |
|---|---|---|---|---|---|
| **Group 1** | 20/40 | 0.227 (0.159) | 0.073 (0.036) | 100 | 0.92 (0.91) |
| [**Group 1** | 22/42 | 0.206 (0.156) | 0.077 (0.038) | 117 | 1.07 (1.07)] |
| **Group 2 *** | 9/23 | 0.255 (0.159) | 0.067 (0.034) | 82 | 0.90 (0.97) |
| [**Group 2** | 11/25 | 0.201 (0.149) | 0.088 (0.044) | 138 | 1.73 (4.21)] |
| **Group 3 *** | 7/17 | 0.280 (0.152) | 0.081 (0.042) | 90 | 2.33 (1.41) |
| [**Group 3** | 9/20 | 0.229 (0.149) | 0.106 (0.054) | 145 | 2.33 (1.36)] |

The analyses of population subdivision using analysis of molecular variance (AMOVA) using Arlequin software indicate that the majority of genetic variation at more than 90% is distributed within populations (Table 2a). This pattern is not changed when including those individuals with possible geneflow from other species or putatively introduced by humans. If the analysed distribution range is partitioned into three regions (group 1: Austria; group 2: Bavaria; group 3: Baden–Württemberg/Rhineland–Palatinate) and this substructure is considered for AMOVA, the overall pattern of structuring genetic variation does not change, and less than 10% of the molecular variation is distributed among the three groups. However, a major proportion of molecular variance is found to define populations within groups (22.87% for the reduced dataset) (Table 2b). This result is consistent with STRUCTURE analysis (Figure 3) also indicating at *K* = 5 a higher representation of the yellow genetic cluster in Austria and the grey genetic cluster in Baden–Württemberg. However, similar to the other AMOVA results, when group 2 and group 3 are combined and contrasted with group 1, AMOVA partitions only 8.75% of molecular variation among these two regions.

The results from principal component analysis are shown in Figure 4A–C. The three separate analyses followed the result and network structure obtained from SplitsTree (Figure 2). The dataset comprising the same samples used for SplitsTree resulted in a PCA graph separating *G. illyricus* from *G. palustris* and also indicating populations G29 and G30 as outliers (Figure 4A). With the second PCA, the outgroups and outlier G29 and G30 have been excluded, and as a result the five remaining outlier (populations G16, G25, G26; G28 and G19–26) are significantly set apart from the *G. palustris* (Figure 4B). The final analysis, which excluded outgroups and outlier, shows a weak signal to group individuals and populations into distinct groups (Figure 4C). Among the three PCA analyses, variance components explaining the total variance decrease rapidly from 43.3% (A), 17.3% (B) to 11.7% in the final dataset (C). This result is also congruent to AMOVA results partitioning less than 10% of the molecular variation among the three groups (Table 2).

**Table 2.** AMOVA analysis of the distribution of genetic variation (**a**) without assumptions regarding regional structure and (**b**) implying regional substructure (Figures 3 and 4) and separating group 1 (Austria), group 2 (Bavaria), and group 3 (BW and Rhineland–Palatinate). Based on permutation tests in Arlequin, in all cases observed variance component (Va, Vb, and Vc) values and $F_{ST}$ were significantly different from random values. All analyses were performed with the total *G. palustris* dataset (non-reduced) and the reduced dataset (excluding G16, G25, G26, G29, G30, G19#82).

| | Source of Variation | d.f. | Sum of Squares | Variance Components | Percentage of Variation |
|---|---|---|---|---|---|
| (**a**) | | | | | |
| reduced | among populations | 2 | 81.034 | 1.171 (Va) | 9.34 |
| non-reduced | | 2 | 90.838 | 1.158 (Va) | 7.84 |
| reduced | within populations | 77 | 875.341 | 11.368 (Vb) | 90.66 |
| non-reduced | | 84 | 1143.231 | 13.609 (Vb) | 92.16 |
| (**b**) | | | | | |
| reduced | among groups | 2 | 81.034 | 0.858 (Va) | 6.87 |
| non-reduced | | 2 | 90.838 | 0.727 (Va) | 4.94 |
| reduced | among populations | 33 | 488.930 | 2.858 (Vb) | 22.87 |
| non-reduced | within groups | 39 | 721.670 | 4.624 (Vb) | 31.42 |
| reduced | within populations | 44 | 386.441 | 8.782 (Vc) | 70.26 |
| non-reduced | | 45 | 421.561 | 9.368 (Vc) | 63.64 |

*3.3. Phylogenetic Inference, Taxon Identity and Geographic Distribution*

We obtained identical ITS consensus sequences (assembled forward and reverse sequences) for clones and directly sequenced PCR products for *G. palustris* populations G16, G17, G19, G25, G26, G28, G29, and ZWU. For G30, direct sequencing failed and three DNA sequences from PCR clones were obtained and included herein. The direct sequenced PCR products of *G. illyricus* (G22, G23, G24) were assembled from respective forward and reverse sequence data but revealed a number of ambiguities. Sequences based on cloned PCR fragments provided 2, 2, and 3 different clones, respectively. Direct sequencing of *G. italicus* failed, and three different cloned PCR fragments were obtained. In total, 24 ITS sequences were generated (Genbank accession numbers OQ847853–OQ847876).

For phylogenetic ML analysis, additional ITS sequence information was retrieved from Genbank für *G. imbricatus*, *G. palustris*, *G. palustris* x *imbricatus* [*G. xsulistrovicus*], and *G. communis*, which served as outgroup species. These sequences were presented earlier: MK005888-MK005919 [19], KP027306-LP027328 [15], and MF543717-MF543720 [43]. The alignment comprises a total of 83 sequences and a length of 744 bp, including ITS1, 5.8 S rDNA gene, and ITS2 (File S1).

The resulting phylogenetic tree is shown in Figure 5 and divides *G. palustris* into two clades (A1 and A2). All of our herein investigated *G. palustris* individuals group into clade A1 with the exception of one clone from eastern Austrian population G30 showing an intermediate position. The phylogenetic tree obtained from Bayesian inferences (BI) is congruent to the ML analysis, and the tree is shown in Figure S2. Bayesian support values for most nodes are higher compared to ML analysis. This is in particular true for those nodes separating all major clades. BI support values are provided in Figure 5.

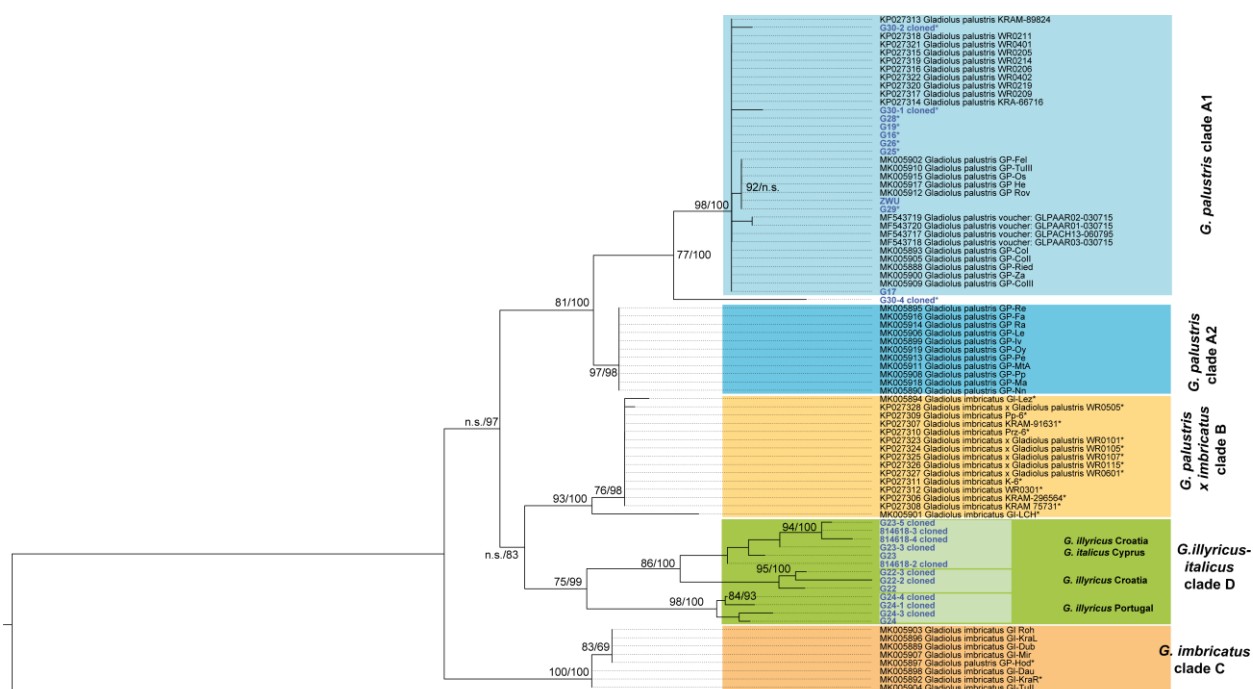

**Figure 5.** Maximum likelihood tree of ITS DNA sequence information. Bootstrap support from ML and Bayesian posterior support from Bayesian analysis (%) is indicated along branches. n.s. denotes bootstrap support <50%. *Gladiolus communis* served as outgroup. Sequences obtained from cloned PCR fragments are labelled as "cloned". Herein newly added sequences are marked in blue. Those selected new accessions that showed an unusual genetic assignment in STRUCTURE analyses are marked with an asterisk.

*3.4. Genetic Assignment of Putatively Introduced Populations by Humans in Baden–Württemberg during the Last 30 Years*

Accessions being assumed to represent introduced and locally non-native populations are G09, G10; G11, G12, G15, and G16 (Supplementary Table S1). However, their individual genetic fingerprints ($K = 5$) are well represented, with the largest native populations in Baden–Württemberg at Wollmatinger Ried near Konstanz (G17) (Figures 1 and 3). Floristic inventarisation of the past decades is documented well, and this would mean that any present *G. palustris* population in the Upper Rhine Valley has been introduced by humans since the early 1990s, as indicated in various personal documentations (Supplementary Table S1). Floristic recording is largely centralized in Baden–Württemberg, and observation data, even including those dating further back than 1900, are accessible via http://www. florabw.recorder-d.de/ [45]. From those, it is obvious that *G. palustris* was not recorded in the Upper Rhine Valley in Baden–Württemberg between 1900 and the late 1980s/early 1990s. However, the species has been shown to be native to this area from a few records from the 19th century (see https://www.floraweb.de/webkarten/karte.html?taxnr=2706; accessed on 5 October 2023; BfN—Bundesamt für Naturschutz, Bonn, Germany). Interestingly, along the Upper Rhine Valley, field observation data have been also provided for *G. imbricatus* (ordnance survey map number 8211/3 from 2014 and 2015), *G. illyricus* (ordnance survey map number 8311/1 from 2004), and *G. communis* (ordnance survey map number 7412/4 from 1973). However, populations have not been confirmed later, and, likely, species determination in the field was not correct. For individuals with AFLP fingerprints available, the comparison of the set of putatively introduced individuals and native populations from Wollmatinger Ried near Konstanz are shown, and both gene pools are congruent based on AFLP genotyping and STRUCTURE analysis (Figure 6).

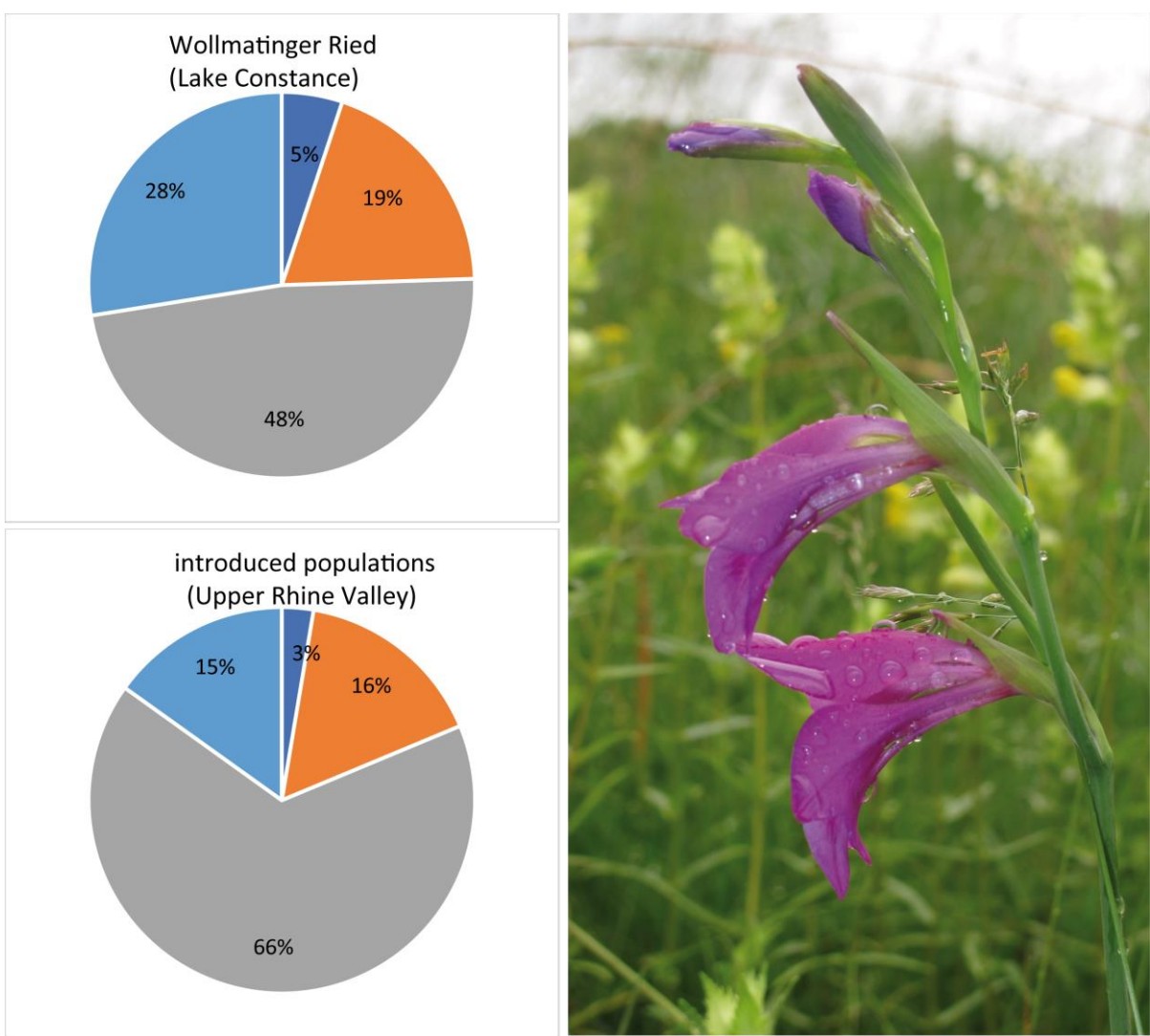

**Figure 6.** Results from genetic assignment (STRUCTURE analysis with *K* = 5) comparing putatively introduced populations of *G. palustris* along the Upper Rhine Valley in Baden–Württemberg (populations G09, G10, G11, G12, G15, G16) with the genetic set-up of the largest native population at Wollmatinger Ried (population 17) near Lake Constance. The image shows flowers from *G. palustris* from an introduced population (G11) (©M.A. Koch, 6 June 2020).

## 4. Discussion

### 4.1. Gladiolus in Central Europe Is a Reticulate Species Complex

Phylogenetic analysis of the ITS DNA sequence variation reveals that the herein studied southern German populations of *G. palustris* belong to a distinct and well-defined clade (clade A1; Figure 4) showing no apparent signs of ancient or recent introgression and hybridization among *Gladiolus* species, although the majority of populations of *G. palustris* studied herein with ITS showed severe signatures of genetic contribution from unknown gene pools based on the results from AFLP data analyses (Figures 2 and 4). So far, the herein presented data do not allow for any further firm conclusions, and detailed multilocus genetic analyses on population level may help to unravel the history of the gene pool of *G. palustris* populations carrying ITS sequence types from clade A1.

Another surprising result is the position of *G. imbricatus* within two different clades. All sequences belonging to clade C have been studied earlier [19]. Clade B combines all remaining accessions of *G. imbricatus*, among them two populations proven genetically to be hybrids of *G. palustris* and *G. imbricatus* [19]. All other accessions from clade B are believed to represent either *G. imbricatus* from Poland and Czech Republic or the hybrid

(Poland) [18]. Bootstrap support for any of these clades is very high. This phylogenetic result indicates that all *G. imbricatus* accessions from clade B may represent hybrids or backcrosses and calls into question the taxon identity presented earlier [18].

In an earlier study [19], a group of accessions was identified that coincides with herein presented clade A2, geographically restricted to a Swiss/French region (French Jura, Ain, Haute-Savoie). These accessions were not recognized as an intermediate gene pool, unlike other more evident genetic hybrids in France and Poland (clade B in Figure 4; *G. palustris* x *G. imbricatus*) [16]. The question arises if *G. palustris* from clade A2 may also represent a group which originated from a secondary contact and hybrid origin between clade A1 *G. palustris* and other populations (e.g., clade B, clade C). In support of this idea, the first evidence comes from the high BI posterior and moderate ML support values indicating a distinct status of clade A2. Furthermore, the geographic distribution of the clade A2 ITS sequence types is restricted (Figure 7). All clade A2 *G. palustris* occur in western Switzerland close to either *G. palustris* from clade A1, true *G. imbricatus* from clade C, or hybrids from clade B, and clade A2 sequences have not been found elsewhere. Vice versa, in Poland and the Czech Republic, *G. palustris* from clade A1, *G. imbricatus* from clade C, and clade B hybrids have been presented. The situation may be even more complex, since two French accessions have been classified as hybrids [16] (Figure 5), one of them carrying a *G. communis* plastid type and the second carrying a plastid type neither assigned to *G. communis*, nor to *G. palustris*, nor to *G. imbricatus*. Here it remains an open question if taxa from the *G. illyricus/G. italicus* clade may have contributed to this genetic signature.

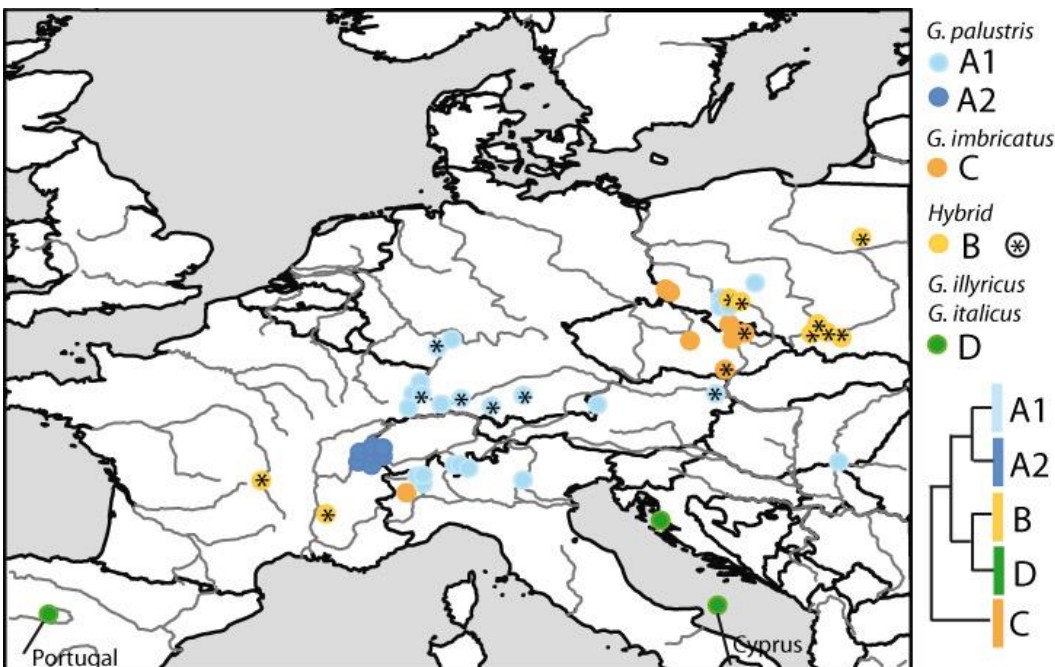

**Figure 7.** Distribution of ITS type variants as inferred by phylogenetic analysis (Figure 5). Hybrids (clades B and C [18,19]) and putatively introgressed individuals (clade A1) are indicated with asterisks.

It is important to note that the ITS region may not be the most optimal marker system to study hybridization and introgression. ITS is part of the rDNA loci and serves as a classical example of concerted evolution of tandemly repeated gene families with a varying number of loci distributed in the genome of plants. The phenomenon of concerted/non-concerted evolution of combined parental ITS types, distributed among those loci within a given genome, makes it challenging to predict the combinatorial outcomes in the following generations. Technically, the study often requires cloning strategies to differentiate within-individual ITS sequence variation [46]. However, in case of *G. palustris,* additional evidence for such reticulate evolutionary processes has been provided via maternally inherited

plastid DNA markers [19]. In Upper Austria, there were no signatures of past hybridization or reticulation detected between *G. palustris* and other species using AFLP data [21].

Within Germany, populations of *G. palustris* did not show significant hybrid signals based on ITS sequence variation. However, multilocus AFLP data provide such evidence of footprints of past genetic contact (denoted with asterisks in clade A1 accessions in Figure 5). In the SplitsTree analysis conducted herein, a few individuals from German populations stood out, displaying notably increased genetic distance compared to all other *G. palustris* individuals (Figure 2). Despite their distinct genetic characteristics, the ITS data clearly placed them within ITS clade A1, leaving the genetic origin of this additional variation unclear. However, one sequenced ITS clone (G30-4) from a Bavarian population appeared to bridge the gap between clades A1 and A2, possibly suggesting little genetic connectivity between these clades. Consequently, when mapping these outlier individuals assigned to ITS clade A1, a genetic connection appears to extend from western Baden–Württemberg and Bavaria towards eastern Austria. Regarding AFLP data, a gradual change in the gene pool structure can be observed from the Upper Rhine Valley towards Bavaria and Austria (Figures 3 and 4C). However, it is worth noting that most of the genetic variation is distributed within populations and regional groups (Table 2), regardless of whether the outlier individuals are included or excluded. This finding is consistent with AFLP data focused on Austrian populations, which also indicated that the majority of the genetic variation exists within and among populations rather than among defined regional groups [21]. Additionally, a slight but significant correlation between genetic and geographic distances was found within the 0–350 km scale [21].

A significant finding of this study from a morphological point of view, particularly when considering data from Poland [18], is that clade B exclusively comprises hybrids of *G. palustris* x *G. imbricatus*, although morphological characters classified them as *G. imbricatus* [18]. This indicates that morphological variations do not necessarily indicate intermediate morphotypes [47,48].

In summary, we can deduce that past introgression and secondary contact among gene pools are more widespread than previously shown [18,19]. A complex postglacial evolutionary history, possibly influenced by human activities over the centuries, has given rise to a mosaic of distinct genetic entities. Moreover, the present-day regional gene pools demonstrate a genetic connectivity, as depicted in the STRUCTURE analysis (Figure 3), and shallow genetic distinctness. This is also seen with the approximation of optimal *K* in genetic assignment of AFLP data. The optimal *K* = 2 result obtained with inclusion of hybrids and putatively introgressed individuals changes to *K* = 5 when those individuals are removed (Table S3A–D). Therefore, it is important to notice that, based on *K* = 5, the shallow signal of regional gene pools is visible on larger geographical scales (Figures 3 and 4).

### 4.2. Hybrids, Introgressed and Also Introduced Populations of G. palustris Should Be Considered Targets of Species Conservation Efforts

Hybridization and introgression are regarded as significant threats to species and biodiversity conservation. It has been argued that the lack of well-defined strategies and guidelines [49,50] exacerbates the challenge. Nonetheless, it is essential to recognize that hybridization and introgression are natural evolutionary processes that have contributed to the formation of species and species complexes [6,7]. The complexities arise when human-induced translocations of organisms and extensive habitat modifications lead to increased hybridization and subsequent introgression, creating serious problems for conservation efforts [15,51]. These factors make it particularly difficult to establish effective conservation strategies and guidelines.

In the case of *Gladiolus palustris* and its likely natural hybrid *G. palustris* x *imbricatus* [=*G. sulistrovicus*], all taxa, including *G. imbricatus*, are rare and experiencing a decline [52–55]. Reticulate evolution has been identified as a major factor in the diversification of gladiolus species [56]. Moreover, postglacial distribution range expansions have led to significant species overlap among European *Gladiolus* species, not only resulting in

natural hybridization [18,55] but also making morphological distinction challenging due to overlapping ranges of morphological characters [47,48].

The herein studied small populations are typically found in ecologically suitable but often isolated habitats. These habitats are rich in biodiversity and enjoy legal protection. It is important to note that none of the studied local populations currently exhibit ongoing hybridization and introgression between different genetic entities, gene pools, or species because of severe geographical and habitat isolation. Populations G09, G10, G12, and G15 were recorded for the first times in 1994, 1995, 1997, and 2001, respectively. All populations are within the same area along the Upper Rhine River (Figure 1). Populations G11 and G16 were monitored for the first time in 2007 and 2012, respectively. From all these introduced populations, population G16, which is geographically close to population G15, is a genetic outlier (Figures 2 and 4B), and at the same time, it is the most recently introduced population recorded. The introduction history of population G30 is uncertain, and it is believed to have been growing at this site since 1977 (Table S1). This population is one out of the two extreme genetic outliers (Figures 2 and 4A). Population size of all introduced populations is small (<100 individuals) with G12 (c. 85 individuals), G11 (c. 60 individuals) and G16 (c. 27 individuals) representing populations of moderate size. All remaining populations are represented by less than 10 individuals each (Table S2).

Given the context of introduced populations along the Upper Rhine River, a question arises regarding their status as targets for species and habitat protection. It is shown that putatively introduced *G. palustris* populations build up a coherent gene pool along the Upper Rhine Valley, which corresponds well with the overall gene pool in Rhineland–Palatinate and Baden–Württemberg. As such, several factors support their consideration for protection: (i) None of these populations poses a severe threat to the genetic identity of established populations; (ii) all populations adequately represent the regional gene pool; (iii) the regional gene pool is reflective of a larger-scale differentiation (see Figures 3 and 5); and (iv) the local southwestern German gene pool (group 3) contributes significantly to overall genetic diversity (refer to Table 1).

As a result, it can be concluded that not only naturally occurring hybrids and introgressed populations but also introduced populations should be regarded as targets for species protection [5,16,57,58], necessitating corresponding monitoring and management activities. Because all newly introduced populations are small and isolated, efforts should be made to increase population size, minimise negative effects such as genetic drift, and adopt management plans accordingly.

**Supplementary Materials:** The following supporting information can be downloaded at: https://www.mdpi.com/article/10.3390/d15101068/s1, Figure S1: Genetic population structure of *G. palustris*; Figure S2: Bayesian inference of phylogenetic relationships; Table S1: Accession list; Table S2: AFLP data matrix; Table S3: Posterior probabilities for estimated numbers of genetic clusters; File S1: ITS alignment.

**Funding:** This research received financial support from Heidelberg University, Marsilius-College, in 2022/2023 granting a fellowship to the author (MAK) under the working project title: Dangerous and Endangered Landscapes. For the publication fee we acknowledge financial support by Deutsche Forschungsgemeinschaft within the funding programme "Open Access Publikationskosten" as well as by Heidelberg University.

**Institutional Review Board Statement:** Not applicable.

**Data Availability Statement:** The data supporting the results of the presented study are available via supplementary materials to be downloaded from the publisher's homepage. DNA sequence data have been deposited with GenBank.

**Acknowledgments:** I thank Peter Thomas and collaborators for providing samples of leaf material from Germany and France. Special thanks go to Andreas Tribsch (Salzburg) for sharing DNA used in earlier studies, to Lisa Kretz and Anna Loreth for wet-lab work, to Peter Sack for collection management, and to Katharina Orth for support with data curation.

**Conflicts of Interest:** The author declare no conflict of interest.

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
