# Peer review of "The Diverse Reticulate Genetic Set-Up of Endangered Gladiolus palustris in Southern Germany Has Consequences for the Development of Conservation Strategies"

_diversity, doi:10.3390/d15101068_

Round 1
Reviewer 1 Report
The manuscript definitively needs improvement in terms of orthography and punctuation (see my comments of the English language). Without these improvements, the manuscript will make a very superficial and sloppy impression despite its scientific soundness and noteworthiness.
Throughout the text:
Lake Konstanz = Lake Constance
Rheinland-Pfalz = Rhineland-Palatinate
do not use abbreviations in the text (l. 265: no. = number; l. 268: pop. = population)
man = humans
improve punctuation [e.g., l. 296: "In total, 24 ITS sequences (...)"]
asteric = asterisk
check verbs in sentences (e.g., l. 482: "then the intermediate phylogenetic position of A2 becomes noteworthy")
scientific names should be put into italics (e.g., l. 498, 514, 515, 539)
avoid colloquial expressions (l. 528: "it´s")
make complete sentences (l. 283-285)
Author Response
Please see the attachmentReviewer 2 Report
General comments
Marcus Koch provides solid analytical work on the genetic-setup of the highly endangered Sword-Lily (Gladiolus palustris) in Central Europe. The focus of the study was to characterize the major gene pools of the species in Southern southwestern Germany using AFLP markers and ITS DNA sequencing. The research showed that past hybridization and introgression are more extensive than previously thought. The status of introduced populations in the German federal state of Baden-Württemberg was also analyzed. Koch suggests promoting and protecting these new populations as they have been present since a long time in that region. He also proposes that naturally occurring hybrids and introgressed populations of the highly endangered G. palustris should be a primary target of conservation efforts.
The understanding of the effects of hybridization and introgression of closely related rare plant species, is essential to the conservation of long-lived rare plants but there are only a few studies that analyzed the genetics of such populations. The present study underpins the importance to consider also introduced populations in the conservation of rare plant species.
The questions addressed are interesting and the manuscript is generally well written but the results and discussion part and the figure and table captions need some rewriting and maybe some new interpretations. A part of the results section could be moved to the discussion and the text could be streamlined.
There are also some specific comments and suggestions.
Title
The title is not very clear. Why should the reticulate genetic setup ‘impede’ conservation strategies?
Abstract
Page 1 line 12: Change ‘...mean temperatures….’ to ….‘mean annual temperatures
Page 1 lines 13-14: Rephrase ‘...genetic depletion with other species…’. It is not clear what is meant here.
Page 1 line 16: Move the phrase ‘The upper Rhine valley…’ down to line 21 before the phrase ‘Introduced populations….
Page 1 line 18: Change ‘... the research…’ to ‘ … this study….’.
Page 1 line 30: Rephrase ‘...plant species in Central Europe are more and more endangered, irrespectively if they…’.
Page 1 line 34: Delete ‘...are manifold and….’
Page 1 line 36: Use past simple, change ‘...has..’ to ‘...had…’.
Page 1 lines 36-37: Change ‘...severe impact while changing temperature…’ to ‘...severe impact with changing temperature…’
Page 1 line 38: Change ‘.....isolation and dispersal limitations is…’ to Change ‘.....isolation and dispersal limitations are…’
Page 1 line 39: Change ‘ …inbreeding depression and loss of adaptive capacity is …’ to ‘ …inbreeding depression and loss of adaptive capacity are …’
Page 2 lines 48-49: Rephrase the sentence, it is not very clear.
Page 2 line 57: Change ‘....among closely related taxa or merging’ to ‘....among closely related taxa or the merging….’
Page 2 line 60: Rephrase ‘Among such biological examples….’
Page 2 line 66: Add reference
Page 2 lines 68-72: Delete sentences ‘It was taxonomically classified…Gladiolus triphyllos Bertol.’
Page 2 line 74: Change ‘In the case of G. imbricatus naturally occurring hybrid…’ to ‘In the case of G. imbricatus, a naturally occurring hybrid…’
Page 2 line 81: Change ‘However, finally…’ to ‘However,...’
Page 3 line 85: Change ‘ However, during past decades few scattered…’ to ‘During the past decades a few scattered….’
Page 3 line 87: ‘.. introduced by man..’ to ‘...introduced by humans..’. Use human instead of man throughout the text. Another possibility would be to say simply ‘introduced’ as ‘by humans’ is implied.
Page 3 line 90: Change ’...a representative set of populations from Western Europe have…’ to ’...a representative set of populations from Western Europe has…’
Page 3 lines 91-93: Move this part to methods.
Page 3 lines 94-97: Rephrase this paragraph, wording is not clear.
Material and Methods
Page 3 lines 103-104: Rephrase ‘...has been proven to have established…’
Page 3 line 108. Is ‘eventually’ used here in its proper meaning e.g. ‘finally’ ?
Page 3 lines 18-21: Rephrase, text unclear.
Page 3 line 116: Change ‘....have been collected’ to ‘...had been collected…’
Page 4 line 124: Change ‘....introduced by human…’ to ‘....introduced by humans…’ Use humans consistently throughout the text. At present there is a mix of ‘man’ and ‘human’.
Page 4 line 126: Change to ‘ For the AFLP analysis 42 locations had been sampled to represent G. palustris in the study region (native or introduced populations),....
Page 4 line 134: How many G. illyricus samples were used?
Page 5 line 169: What do you mean by ‘..the tree-selective PCR products….’?
Page 5 line 189: Why did the STRUCTURE analysis only consider K 1 to 5 and not a higher subdivision?
Page 5 line 196: How were the geographical regions defined? The STRUCTURE analysis indicated K=2, or K=5? See also Figure 3. Explain.
Page 5 line 203: Replace ‘pops’ by populations throughout the text.
Page 5 lines 204-205: Rephrase, text unclear.
Page 5 line 207: Explain why cloning was used for some individuals and not for others?
Page 6 lines 211-212: Explain why this is interesting?
Results
Some parts of the results section already contain conclusions. These parts should be moved to the discussion. The text also needs some stream lining.
Page 6 lines 241-242: Change to ‘..good evidence for the genetic separation of G. illyricus…’
Page 6 lines 247-248: Change to ‘STRUCTURE analyses of the same G. palustris dataset…’
Page 7 line 250: In the discussion -> reconsider the result that the ‘optimal K was 2’ -> see ‘The K=2 conundrum’ in Janes et al. 2017
Page 7 lines 261-264: Explain what is meant here: ‘Inclusion of those outlier individuals increased regional diversity consistently, ….; and - as a consequence - the estimated amount of regional expected heterozygosity decreased …’
Page 7 lines 266-267: ‘… (p < 0.01, for any comparison).’ Explain how this result was obtained -> ANOVA + Tukey test? Explain also in materials and methods.
Page 7 line 281 Figure 3: Interpret the results of the STRUCTURE analysis for K = 5.
Page 7 line 284: Change ‘... and contrasts those with group 3 (Austria)..’ to ‘... and contrasts those with group 1 (Austria)..’. Check the numbering of the groups throughout the text.
Page 7 lines 280-286. Difficult to understand what is meant here. These sentences need some rephrasing. Also be more precise and move these sentences to the discussion.
Page 7 line 292: What is meant by sequence types?
Page 8 lines 293-294: Text unclear. Did the ambiguities appear after assembling? Why were these populations assembled?
Page 8 lines 298-301: Move this part to materials & methods. Check the reference number for the accessions -> the MK accessions are from publication 19.
Page 8 lines 304-331 + Figure 4: This part is unclear and should be moved to the discussion. Moreover we question the interpretation of the results of the ML analysis. Nodes with low support (denoted n.s.) should be collapsed. As this may change the position of the different clades a new interpretation is needed. In addition to the ML analysis an analysis by a Bayesian method like Mr. Bayes could give additional support for the tree structure. Moreover a clear identification of new hybrids would need an analysis of cpDNA markers.
Page 8 lines 309-310: Rephrase this sentence. Check the reference number.
Page lines 314-316: What is the evidence that clade A2 includes secondary hybrids between clade A1 and B? Change ‘clade 1’ to ‘clade A1’.
Page line 321: Change C. imbricatus to G. imbricatus.
Page 8 lines 314-316. Move the last sentence to the discussion and elaborate more clearly what is meant here and explain why you think that clade A2 may represent a group of secondary hybrid origin.
Page 8 lines 327-330. ‘...we may rely on AFLP data to identify such footprints of past genetic contact…’. ’I suggest adding an PCoA or a DAPC analysis of the AFLP data to compare the results to the SplitsTree and STRUCTURE analyses.
Page 8 line 329: ‘ …ITS often failed to detect hybridization…’ add reference.
Page 9 lines 335-338. Captions of both Figure 2 and Figure 3 are not clear. Add population G17 to Figure 2. In Figure 3 it is not obvious how the introduced populations are ‘..well represented within the largest native populations… (G17)....’ I suggest either to add the population abbreviations to the STRUCTURE figure or at least -> indicate the introduced populations + population G17. There is also a mistake in the denomination of the groups in the caption of Figure 3. Change ‘..SW Germany (group 3…)...’ to ‘..SW Germany (group 1…)...’ Check for consistency in the text for the group numbering.
Page 9 lines 350-351: It is not clear what is the meaning of the numbers ‘... G. imbricatus (8211/3…) -> are these German geographical coordinates? Explain.
Page 9 lines 355-356. Indicate that this conclusion is based on the STRUCTURE analysis. Again this sentence should be moved to the discussion section.
Figures
Page10 lines 360-364: Explain in the caption what is meant by n=6, n=9, n=4 etc… Change ‘.....individuals have been studied ..’ to ‘...individuals that have been studied….’. Are the ‘populations with unusual placement’ identical to the ‘individuals that have been studied’? Indicate what is the difference between the square symbol and the point symbol.
Page 11 line 384. Considering that a figure legend should always allow you to understand what is shown without reading the main text be more precise (this concerns all figures). Indicate what species had been analyzed, how many populations etc… Add population G17 to the figure as it is referred to in the text. Explain that the oval in the middle indicates potential hybrid individuals.
Page 12 line 391: As already mentioned -> there is some confusion with the group numbering, please double check. Write population names under the barplot. Explain which pie chart corresponds to which region. Why is the upper bar chart divided by country when otherwise the three regions are indicated? Be consistent.
Page 13 lines 404-405: Place the bootstrap values near the nodes. Be more precise in the legend what data are shown in the figure p.ex. the potentially introduced populations, the outgroup etc…
Page 13 lines 410-412: Change ‘...as documented with…’ to ‘ …as inferred by..’. What do you mean by ‘proven hybrids’?
Page 14 Figure 6. Use population codes in the pie charts or in the legend. P.ex. G17 = Wollmatinger Ried. Indicate that the photograph is showing a plant form an introduced population.
Page 15 line 432: Explain what is meant by ‘..three regions studied (group 1,2,3;..’. Indicate why some numbers are in bold.
Page 15 lines 447-448: Check if the group numbers are consistent with the text.
Page 15 lines 455-468: Explain what is meant by Va, Vb, Vc?
Discussion
The discussion has to be rewritten by integrating some parts of the results.
Page 16 lines 475-476: Explain why there are no apparent signs of hybridization if some individuals of clade A1 were indicated as hybrids in the AFLP analysis in Figure 2?
Page 16 lines 478-485: This part is difficult to understand and speculative. Why should clade A2 have originated from a secondary contact? Focus on the position of your new ITS sequences in clade A1 in comparison to the AFLP analysis.
Page 16 lines 511-512: ‘...a slight but significant correlation between genetic and geographic distances was found…’ -> Isolation by distance. Where do these results come from? Describe the analysis in the methods part.
Page 17 lines 518-519: Add reference to ‘In summary, we can deduce that past introgression and secondary contact among gene pools are more widespread than previously believed.’
Page 17 line 522: Be more precise ‘... as depicted in the STRUCTURE analysis.’
Page 17 lines 528-530. Indicate a reference
Page 17 line 552: Check again for consistency with the group numbers.
Page 18 lines 554-556: Indicate some references. Maybe add also a sentence stating that in order to avoid the negative effects of genetic drift, population sizes of all small populations (including those introduced by humans) should be increased. ‘It is important to note that none of the studied local populations currently exhibit ongoing hybridization and introgression between different genetic entities, gene pools, or species.’ Provide some evidence for this statement.
The text should be revised by a native English speaker.
Reviewer 3 Report
Abstract
Rows 18-19: the research shows that past hybridization and introgression in Central Europe are more extensive than previously thought, phrase is not logistic. [the same concerns discussion – „In summary, we can deduce that past introgression and secondary contact among gene pools are more widespread than previously believed“]
Furthermore, the research proposes that naturally occurring hybrids and introgressed populations should also be the primary target of conservation efforts. Use of expression introgressed populations
is under question. Better information for the abstract would be author research based facts which pairs of Gladiolus species were detected forming natural hybrids and which pairs of Gladiolus species are forming hydrids with Gladiolus palustre
Abstract does not contain results of the study, all assumptions should be as supplementary to the data obtained.
Comparison of AFLP and ITS data is missing
Keywords: some words are replicating title, some possible another keywords might extend probability of citations
Lowercase letters should be used for English names of Gladiolus species. Gladiolus palustris (marsh Gladiolus)... In Germany Marsh Gladiolus..
marsh Gladiolus nomenclature, either common English or international and titles should be unique throughout the paper– admixture of international and English names is not the best practice for the binomial. The best would be to use international names as they are provided in some figures, in all text italic letters for international names should be used.
All data about registration in the gene bank could be placed in the method chapter.
Terminology
It is not clear what the author names as populations and what as accessions.
Fig. 6. native population „system“ - what is assumed as popolation system?
Inadequate titles of the circles
In some places word populations and in some other accessions are used
„As outgroup accessions three G. illyricus populations“ Do accession and population have the same meaning?
Mess in geographical titles throughout the manuscript, including tables, figures and plain text.
SW-Germany, S-Germany and Austria
Germany, Austria
SW Germany (group3, Baden-Württemberg and Rheinland-Pfalz); blue circle: S Germany (group2, Bavaria plus two populations from western Austria); orange circle: Austrian populations (group3).
Unification is required. In addition, one more type of names “group 1, 2 3” (these titles are meaningless) is very confusing.
Lake Konstanz populations – have they been examined and why this lake is not shown in the map near the lake Konstanz.
Mess in colours throughout the manuscript – one color should have the same meaning along all the paper. Advice should be to remove colors from the locations (circle, ellipses) in the map (Fig. 3).
Fig. 3 and Fig. 4-5 should have colors of very distinct palette
Fig. 2. Title is not informative.
Fig. 2 provides the third colour system. It is not clear to which coloured dots G09 and G15 belongs
Some acronyms are coloured and some not. Some lines without cirsles and some with.
Fig. 3
x – three parts of the compound figure should have labels
x – bar plot should have vertical lines separating populations, each population should have name/acronym/abbreviation in that bar plot like it is shown in the Fig.
x - bar plot / ribbon – titles of the countries – according to such geography the article is devoted to explain differences between the countries only;
x - discs of gene clusters are not explained in the notes;
x - fragments of the bar plot in the map should be connected by dotted lines with the entire bar plot of all investigated populations. Red colour is used in two meaning – sites and gene clusters.
x - no explanation for red circles and blue quadrangles.
x - SW Germany (group3, Baden-Württemberg and Rheinland-Pfalz); blue circle: S Germany (group2, Bavaria plus two populations from western Austria); orange circle: Austrian populations (group3).
Some data references and comments (needs and dates) for the populations with putative introgression are required.
Fig. 6. Two cases of gene clustering is provided although beside only one photo is provided. It might be better to title the photo by separate figure or remove at all.
Groups in the AMOVA table should be detailed
Some comments about former introduction of populations of G. palustris
Content of the Suppl. table 1 is not clear and do not have relation to the article material
In the methods information about individuals examined in each population is missing.
No data, which individuals were analysed by AFLP and which by ITS and which populations have been taken from the gene bank.
Round 2
Reviewer 3 Report
In the abstract „The Upper Rhine Valley has seen the emergence of various scattered populations of G. palustris over the past three decades, which are believed to have been introduced by humans. Introduced populations in Baden-Württemberg likely descend from a large source population near Lake KonstanzConstance.“
In addition to well known countries‘ titles all these geographical localities Upper Rhine Valley Baden-Württemberg Lake KonstanzConstance should be demonstrated in the map – for the international reader it is not clear, presented map might be clear for the reviewer from close to the Germany or Austria countries.
All these geographical localities Upper Rhine Valley Baden-Württemberg Lake KonstanzConstance should have population titles in the parantheses otherwise it is not clear what the author is talking about in the abstract.
Letters of populations in the fig.1 should be bigger – there is enough space to increase them- size of the letters should approach the size of the letters in the title of the figure. If Gladiolus imbricatus is titled in the figure it should be also G. palustris written and localized. – No unifirmity
Populations of the different species should have different labels in the map
Coloured dots of the sites in the map of the fig. 2 should be the same in the fig. 1 - in this figure dots are black points.
Fig. 3
Bayesian stripes with populations still missing titles (abbrev.) of the populations (for x axis titles are missing (they might be placed on the top of the strip)
It is not clear from the title what is presented in the first strip and what is presented in the second
Very strange/unusuall order of the clusters in left-right direction – 3 – 2 – 1
Sample from internet: (populations in numbers), in the article under review – population abbreviated titles like in figure should be used
Fig. 4 – letters should be bigger in the left side plots.
G. illyricus (mentioned in the title) is missing in the plots
group 3 BW – letters BW should be replaced by meaningfull words
Another example from internet, in this tree everything is clear – both which populations are presented and which region do they represent
The author is stating that he prefers laconic style. Either laconis, or comprehensive – in all cases it should be clear for the reader.
Remarks for the figures and table (of supplement about the sites) have not been taken into account.
The readers should not spent exsessive time analysing content of the figures.

Round 3
Reviewer 3 Report
Thank you for updates